# SHIV remission in macaques with early treatment initiation and ultra long-lasting antiviral activity

Michele B. Daly[1], Chuong Dinh[1], Angela Holder[1], Donna Rudolph[1], Susan Ruone[1], Alison Swaims-Kohlmeier[1,6], George Khalil[2,7], Sunita Sharma[1], James Mitchell[1], Jillian Condrey[3], Daniel Kim[1], Yi Pan[2], Kelly Curtis[1], Peter Williams[4], William Spreen[5], Walid Heneine[1] & J. Gerardo García-Lerma[1] ✉

Studies in SIV-infected macaques show that the virus reservoir is particularly refractory to conventional suppressive antiretroviral therapy (ART). We posit that optimized ART regimens designed to have robust penetration in tissue reservoirs and long-lasting antiviral activity may be advantageous for HIV or SIV remission. Here we treat macaques infected with RT-SHIV with oral emtricitabine/tenofovir alafenamide and long-acting cabotegravir/rilpivirine without (n = 4) or with (n = 4) the immune activator vesatolimod after the initial onset of viremia. We document full suppression in all animals during treatment (4-12 months) and no virus rebound after treatment discontinuation (1.5-2 years of follow up) despite CD8 + T cell depletion. We show efficient multidrug penetration in virus reservoirs and persisting rilpivirine in plasma for 2 years after the last dose. Our results document a type of virus remission that is achieved through early treatment initiation and provision of ultra long-lasting antiviral activity that persists after treatment cessation.

Antiretroviral therapy (ART) has drastically improved clinical outcomes for people living with HIV. However, ART is not curative and requires lifelong medication adherence to ensure viral suppression and limit progression to AIDS. The inability of ART to cure HIV is due to the rapid establishment of a virus reservoir predominantly in memory CD4 + T cells[1]. The T cell archive of replication-competent virus is the primary source of virus rebound in HIV-1-infected persons who discontinue ART and represents a significant challenge to achieving an HIV cure[2]. Attempts to achieve a functional or sterilizing cure have therefore focused on targeting the latent virus reservoir with various approaches, including gene-editing technology, "shock and kill"

strategies to reactivate and eradicate the latent virus, and latency silencing[3]. In a minority of patients, early ART initiation has been associated with virus remission highlighting the potential of this strategy for HIV cure[4–7].

Macaque models of sexual HIV infection have been used to study the impact of early treatment on reservoir formation and to precisely time reservoir seeding following rectal infection with SIVmac251[8,9]. Whitney et al. used a subcutaneous three-drug ART regimen of tenofovir (TFV) or tenofovir disoproxil fumarate (TDF), emtricitabine (FTC), and dolutegravir (DTG) initiated before or after the onset of acute viremia and continued for 6 months. Despite suppressive ART,

¹Laboratory Branch, Division of HIV Prevention, National Center for HIV, Viral Hepatitis, STD, and TB Prevention, Centers for Disease Control and Prevention, Atlanta, GA, USA. ²Quantitative Sciences and Data Management Branch, Division of HIV Prevention, National Center for HIV, Viral Hepatitis, STD, and TB Prevention, Centers for Disease Control and Prevention, Atlanta, GA, USA. ³Comparative Medicine Branch, Division of Scientific Resources, National Center for Emerging and Zoonotic Infectious Diseases, Centers for Disease Control and Prevention, Atlanta, GA, USA. ⁴Janssen Research & Development, Beerse, Belgium. ⁵ViiV Healthcare, Research Triangle Park, NC, USA. ⁶Present address: Department of Microbiology & Immunology, Emory University School of Medicine, Atlanta, GA, USA. ⁷Present address: Office of Informatics and Data Management, Centers for Disease Control and Prevention, Atlanta, GA, USA. ✉e-mail: jng5@cdc.gov

the virus rebounded after ART discontinuation in all animals that initiated ART on day 3 after infection and before detectable viremia, or on days 7–14 post-infection during acute viremia[8,9]. These data demonstrate that the virus reservoir is seeded very early during the eclipse phase of infection and is particularly refractory to conventional suppressive ART. We posited that disruption of early reservoir formation may require different ART modalities that are more potent and have increased tissue penetration at reservoir sites. We designed a 4-drug ART regimen containing daily oral tenofovir alafenamide (TAF) and FTC, and intramuscular cabotegravir long-acting (CAB LA) and rilpivirine long-acting (RPV LA). TAF was selected over TDF for its improved dosing of lymphoid tissues and peripheral blood mononuclear cells (PBMCs) which results in ~ 6-fold higher concentrations of the pharmacologically active metabolite, TFV-diphosphate (TFV-DP)[10–12]. We administered TAF and FTC orally, as opposed to subcutaneously, to better recapitulate tissue biodistribution in humans[13,14]. CAB and RPV were selected for their added advantage of penetrating the central nervous system[15]. The long-acting formulations of CAB and RPV were chosen over oral formulations because of the long pharmacologic drug tail that persists after treatment which may enhance virus control. In humans, the apparent terminal elimination half-lives for CAB and RPV after a single intramuscular injection are 39–80 days and 91–196 days, respectively[16]. We tested the ability of early treatment with this regimen to achieve virus remission in macaques rectally infected with a pathogenic simian HIV isolate that contains the HIV-1 reverse transcriptase (RT-SHIV). Animals were treated after the onset of viremia for 1 year and monitored after treatment discontinuation for virus rebound and seroconversion for 2 additional years including a period of experimental CD8 + T cell depletion. In a second group of animals that also initiated treatment after the onset of

viremia, we tested if the addition of the immune activator vesatolimod (VES) to a shorter 4-month treatment could also achieve remission. We document durable virus remission (defined as lack of detectable viremia in the absence of treatment) in all animals treated with this optimized ART regimen with or without VES despite CD8 + T cell depletion and detectable intact proviruses 1.5–2 years after treatment cessation.

## Results

### Durable remission with early ART

We infected 4 macaques intrarectally with a pathogenic RT-SHIV that contains the HIV-1 reverse transcriptase (RT)[17–19]. This chimeric virus was chosen because SIV RTs are not susceptible to non-nucleoside RT inhibitors (NNRTIs)[20]. We initiated treatment with daily FTC/TAF and monthly CAB LA/RPV LA after two consecutive SHIV RNA positive measurements in plasma (5 to 6 days after SHIV inoculation) (Fig. 1a). The median (range) plasma RNA levels at treatment initiation were 3.1 (2.3–3.8) $\log_{10}$ copies/mL with peak levels (3.4 (2.7–4.3) $\log_{10}$ copies/mL) seen 10 days post-infection (Fig. 1b). All 4 early ART (eART) treated macaques were virologically suppressed at day 19–28 post-infection and remained undetectable during treatment with FTC/TAF/CAB LA/RPV LA (months 1–6) and maintenance therapy with CAB LA/RPV LA (months 6–12) (Fig. 1c). In contrast, untreated animals (n = 4) infected with the same dose of RT-SHIV had high peak viremias (median = 6.9 (range = 6.7–8.1) $\log_{10}$ copies/mL) and virus set points that ranged between 3 and 7 $\log_{10}$ copies/mL (Fig. 1b). Two of the untreated controls were rapid progressors showing a drastic decline in CD4 + T cells to 44 and 275 cells/μL and progression to simian AIDS within 6 months. The remaining 2 controls had 540 and 1127 cells/μL at the time of necropsy at month 11. After ART discontinuation at month 12, none of

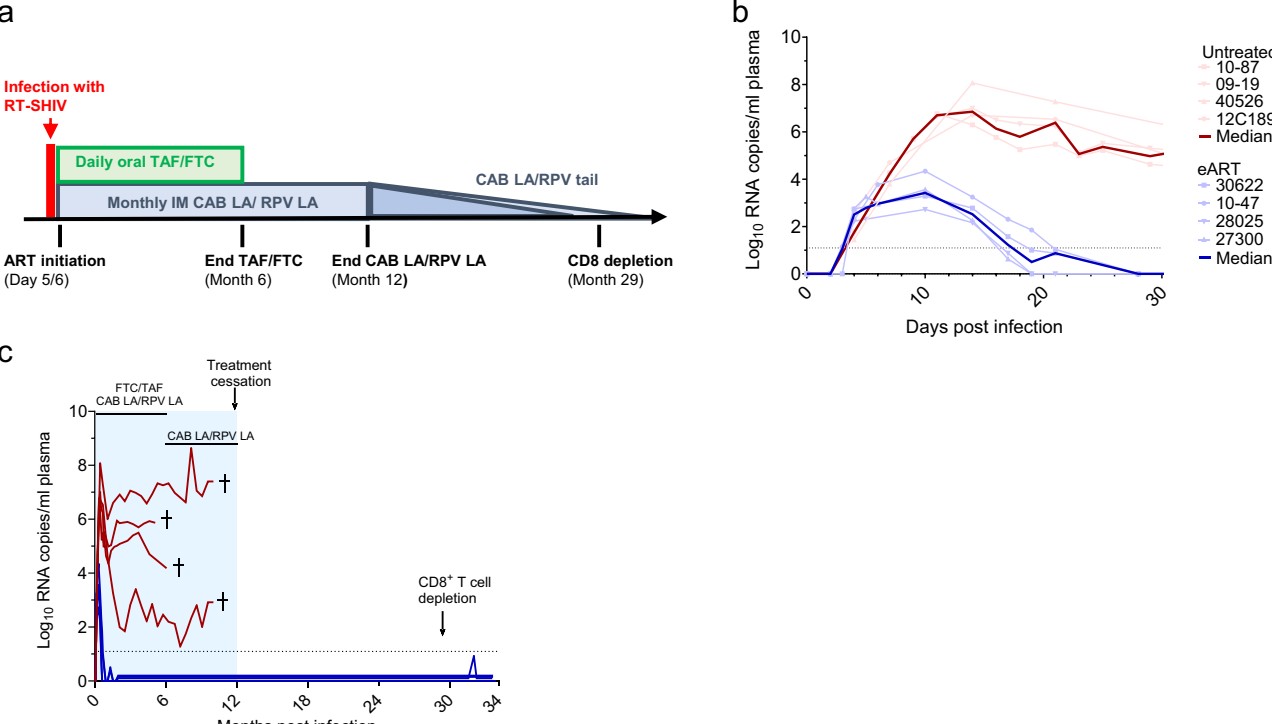

**Fig. 1 | Study design and SHIV infection dynamics. a** Experimental Design. Macaques infected with RT-SHIV (*n* = 4) were treated with daily tenofovir alafenamide/emtricitabine (FTC/TAF) and monthly intramuscular injections of cabotegravir and rilpivirine long-acting (CAB LA/RPV LA) 5 to 6 days after infection. Treatment was continued for six months followed by maintenance with CAB LA/RPV LA until month twelve. At month 30, macaques received an anti-CD8 monoclonal antibody MT807R1 to deplete CD8 + cells. **b** Acute SHIV viremia in treated (blue

lines) and untreated (red lines) macaques. The thick lines denote median values. **c** SHIV RNA levels in treated (blue) and control (red) macaques during treatment (shaded blue area) and after treatment cessation. The crosses indicate the time of euthanasia. The horizontal dotted line denotes the limit of quantification (12.5 copies/mL). ART; antiretroviral therapy. Source data are provided as a Source Data file.

the treated animals experienced virus rebound during a follow-up period that extended for almost 2 years (Fig. 1c).

### Lack of virus rebound after CD8+ T cell depletion

To further understand the degree of immune virus control after treatment cessation we depleted CD8 + T cells in the treated animals 17 months after treatment cessation when CAB was no longer detected in plasma and RPV levels were below plasma levels seen in humans receiving 600 mg of RPV LA every 4 weeks[21]. Treatment with the CD8 depletion antibody MT807R1 effectively depleted >99.9% of CD8 + T cells in blood as illustrated by a decline in CD8 + T cell frequencies from 43.8% (range = 34.2–47.1) at baseline to 0.018% (range = 0.007–0.038) 18 days after MT807R1 treatment (Supplementary Fig. 1). The frequency of CD8 + T cells remained below 1% until day 48-60 post MT807R1 treatment followed by a gradual increase to 9.5% (range = 4.8–13.5). Despite the drastic reduction in CD8 + T cells, no virus rebound was observed in any of the treated animals within 4 months after depletion, however, one macaque (10–47) had a

transient blip below the limit of quantification 3 months after MT807R1 treatment (Fig. 1c). Overall, these results suggest that CD8 + T cells are not the main contributor to the observed virus remission.

### Limited serologic responses during and after treatment cessation

We next investigated antibody responses against gag (p27), pol (p66), env (gp140, gp130, gp41, gp36), and nef (Fig. 2). In 2 untreated control animals, responses to SHIV were characterized by a rapid increase in antibody levels against gp36, gp41, gp140, gp130, and p66 pol. One of the two untreated animals also mounted a measurable response to nef. In contrast, antibody responses in the 4 treated animals were predominantly against gp140. Two of the four treated macaques also showed a limited response to gp130 that waned over time. Antibody titers did not increase in any of the 4 animals after treatment cessation at month 12 except for one animal that showed a short and transient increase in antibody levels to nef at month 12 and another animal that

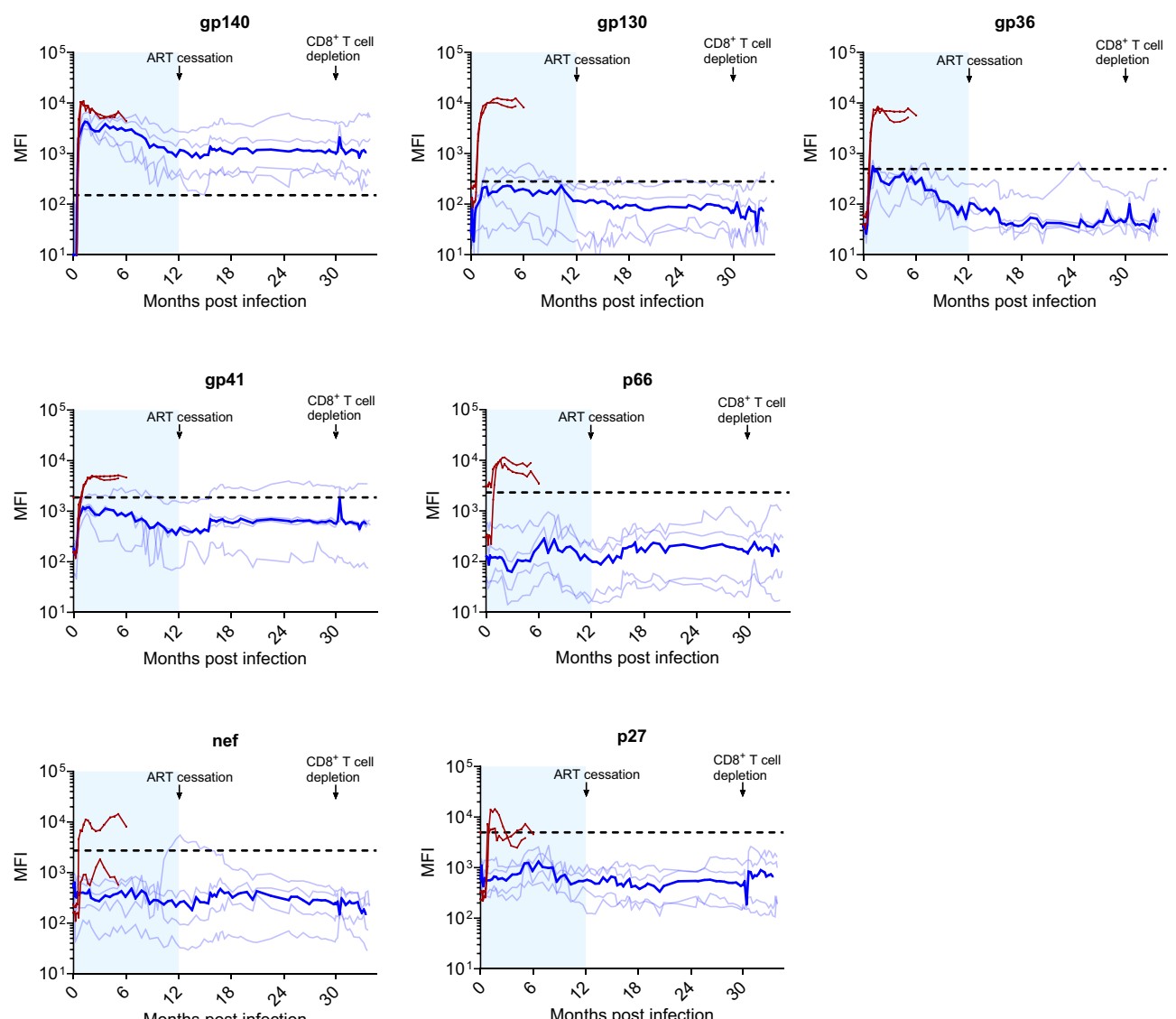

**Fig. 2 | Longitudinal analysis of SHIV antibody levels.** SHIV plasma IgG levels against gp140, gp130, gp36, gp41, p66, nef, and p27 were measured longitudinally using a Bio-Plex assay. Light blue lines denote the mean fluorescent intensity (MFI) for each treated animal; dark blue line indicates the median MFI values. Two untreated controls (macaques 09–19 and 10–87) are shown in red. Treatment in the early ART (eART) animals was initiated 5 to 6 days after infection. The shaded blue area represents the period of treatment with FTC/TAF/CAB LA/RPV LA (months 1-6) and CAB LA/RPV LA (months 6–12). The horizontal dotted line denotes cutoff values for each analyte. Source data are provided as a Source Data file.

showed an increase in the response to gp41 (Fig. 2). Overall, these data reflect very limited and non-evolving SHIV-specific humoral responses.

## Drug concentrations in plasma, PBMCs, and tissues

To better characterize the pharmacologic profile of the ART regimen we measured drug concentrations before and after treatment discontinuation and compared data to those in humans to provide clinical translation of findings. The concentrations of CAB and RPV in plasma and intracellular TFV-DP and FTC-TP in PBMCs are shown in Fig. 3. Median (range) CAB levels during the treatment phase (2400 ng/mL; 830–4980) were 13 times above the protein-adjusted 90% inhibitory concentration (PA-IC$_{90}$ = 166 ng/mL) and within plasma levels seen in humans treated with 400 mg of CAB LA every month for 48 weeks (2740 ng/mL) (Fig. 3a)[21]. After the last injection, CAB concentrations remained above the clinical trough level for treatment of 4 × PA-IC$_{90}$ for 5–6 additional weeks and declined below the PA-IC$_{90}$ two months later (Fig. 3b). The median concentrations of RPV in plasma during treatment was 108 ng/mL (56–182), also within human levels achieved with 600 mg of RPV LA every month for 48 weeks (97.5 ng/mL) (Fig. 3a)[21]. After the last injection, RPV levels slowly declined although they remained about 3 times above the PA-IC$_{90}$ value for RPV (12 ng/mL) for 2 years after treatment cessation (Fig. 3c).

In PBMCs, TFV-DP and FTC-TP concentrations varied between periods of full adherence (months 0–3.5) or suboptimal adherence (months 3.5 to 6) to oral FTC/TAF (Fig. 3d, e). During the period of full adherence to oral FTC/TAF, median TFV-DP levels were 1753 (275–2809) fmols/10$^6$ PBMCs, which is about 3.2 times higher than those observed in humans treated with 25 mg of TAF (525 fmols/10$^6$ PBMCs)[12]. Likewise, median FTC-TP concentrations were 573 (333–1528) fmols/10$^6$ PBMCs, approximately 3.6-fold lower than those observed in humans treated with 300 mg of FTC (2,060 fmols/10$^6$ PBMCs)[12]. During the period of suboptimal adherence to oral FTC/TAF, TFV-DP and FTC-TP concentrations were 329 (63–1688) and 306 (undetectable-964) fmols/10$^6$ PBMCs, respectively (Fig. 3d, e).

We also measured drug concentrations in lymphoid tissue, rectal tissues, and cerebrospinal fluid (CSF) (Table 1). Median concentrations of TFV-DP and FTC-TP in lymph node mononuclear cells (LNMCs) collected from axillary or inguinal lymph nodes were 367 (142–620) and 260 (below limit of quantification (BLQ)–383) fmols/10$^6$ LNMCs, respectively. Median concentrations of TFV-DP and FTC-TP in rectal tissues were 14.6 (5.3–28) fmols/mg of tissue and BLQ (BLQ-17) fmols/mg of tissue, respectively. When the analysis was done by the total amount of extracellular drug per mg of rectal tissue, the levels of FTC, TFV, CAB, and RPV were 1.48, 0.64, 0.60, and 0.45 ng/mg of tissue,

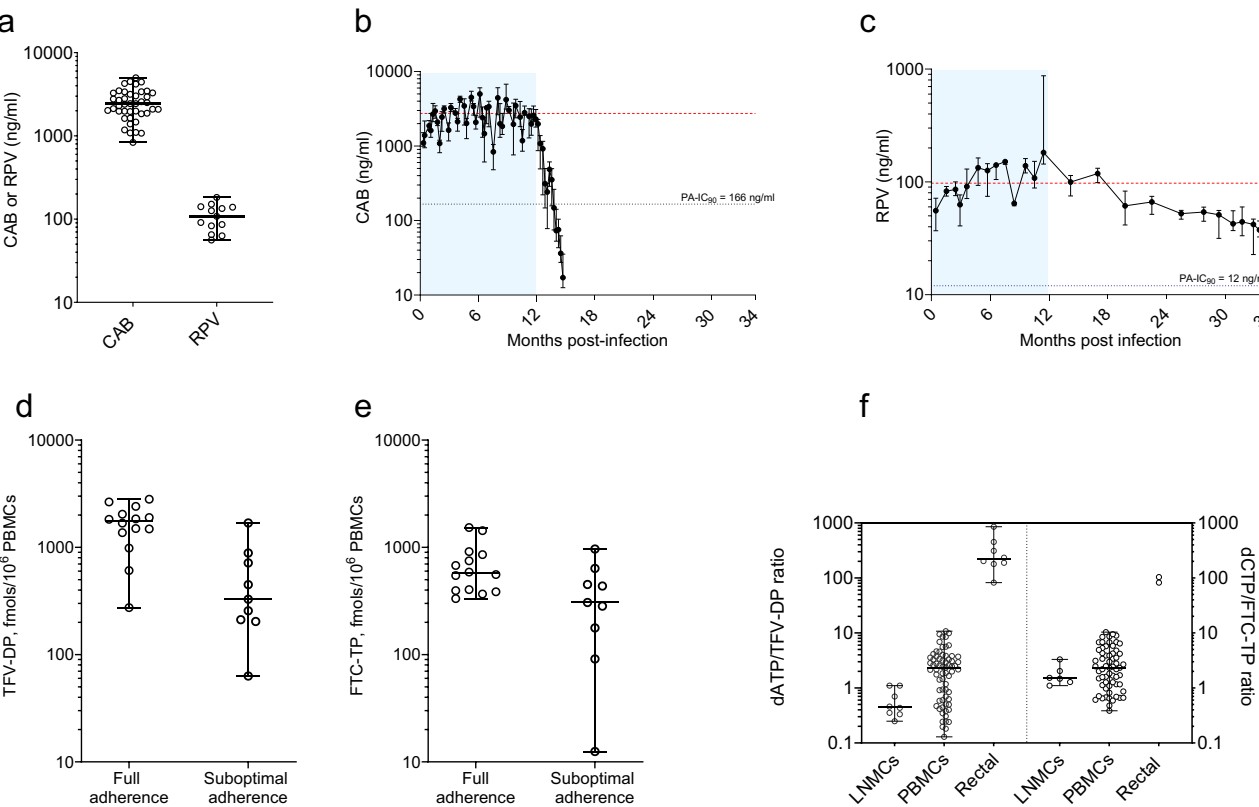

**Fig. 3 | Drug concentrations in blood and tissues. a** Median (range) concentrations of CAB and RPV in plasma during treatment. CAB concentrations were measured every week. RPV concentrations were measured 2–4 weeks after dosing. Each data point denotes the median CAB or RPV levels seen in the 4 treated animals in a given week (42 weeks for CAB and 13 weeks for RPV). **b** Median (range) concentrations of CAB in plasma in the 4 treated macaques during treatment (months 0–12) and after treatment cessation at month 12. The horizontal red line denotes the mean plasma levels seen in humans receiving 400 mg of CAB LA every 4 weeks for 48 weeks. **c** Median (range) concentrations of RPV in plasma in the 4 treated macaques during treatment (months 0–12) and after treatment cessation at month 12. The horizontal red line denotes the mean plasma levels seen in humans receiving 600 mg of RPV LA every 4 weeks for 48 weeks. The shaded blue area in panels b and c denotes the period of treatment. The dotted black lines denote the

protein-adjusted IC$_{90}$ values (PA-IC$_{90}$) for CAB (166 ng/ml) and RPV (12 ng/ml). **d** Median (range) concentrations of TFV-DP in PBMCs measured 24 h post-dosing during the entire study and divided by periods of full adherence to oral FTC/TAF (moths 0–3.5) and periods of suboptimal adherence (months 3.5–6). **e** Median (range) concentrations of FTC-TP 24 post-dosing were also divided by periods of full adherence and suboptimal adherence. Each data point in panels d and e denotes the median TFV-DP or FTC-TP levels seen in the 4 treated macaques in a given week. **f** Ratios between dATP or dCTP and intracellular TFV-DP or FTC-TP in PBMCs, LNMCs, and tissues. Horizontal lines denote medians, and the error bars are ranges. Each LNMC data point denotes ratios obtained in 1–3 biopsies per animal. Each PBMC data point denotes ratios obtained in a given week during the period of full adherence to FTC/TAF. Each rectal data point denotes ratios obtained in 2-3 biopsies per animal. Source data are provided as a Source Data file.

**Table 1 | Drug concentrations in fluids and tissues**

| Analyte | Rectal tissues (fmol/mg or ng/mg)* | Lymph nodes (fmol/10⁶ cells)** | CSF (ng/mL) | Plasma (ng/mL) |
|---|---|---|---|---|
| TFV | 0.64 (0.18–0.87) | n.d. | BLQ | n.d. |
| TFV-DP | 14.6 (5.3–28)* | 367 (142–620) | n.d. | n.d. |
| FTC | 1.48 (0.22–2.16) | n.d. | 2.36 (BLQ-2.65) | n.d. |
| FTC-TP | BLQ (BLQ-17) | 260 (BLQ-383) | n.d. | n.d. |
| CAB | 0.60 (0.25–2.65) | n.d. | 5.95 (4.77–7.43) | 835 (480–1050) |
| RPV | 0.45 (0.12–1.1) | n.d. | 4.06 (2.93–4.52) | 151 (143–153) |

Data are expressed as median (range). Extracellular TFV, FTC, CAB, and RPV concentrations were measured in rectal tissues collected after 1 to 3 months of ART. Intracellular TFV-DP and FTC-TP in rectal tissues and lymph nodes were measured in biopsies collected during the study or an earlier 1-month oral FTC/TAF pharmacokinetic study performed in the same macaques[46]. CSF was collected after ~ 6 months of treatment and 4 weeks after the previous CAB LA/RPV LA injection; data from matched plasma samples are also shown. Source data are provided as a Source Data file.
*fmols/mg of tissue (intracellular TFV-DP and FTC-TP) or ng/mg of tissue (TFV, FTC, CAB and RPV).
**fmols/10⁶ lymph node mononuclear cells.
*TFV* tenofovir, *TFV-DP* tenofovir diphosphate, *FTC* emtricitabine, *FTC-TP* emtricitabine triphosphate, *CAB* cabotegravir, *RPV* rilpivirine, *CSF* cerebrospinal fluid, n.d., not done, *BLQ*: below the limit of quantification.

respectively (Table 1). In CSF, CAB showed the highest concentrations (5.95 ng/mL) followed by RPV (4.06 ng/mL) and FTC (2.36 ng/mL). TFV was not detected in any of the CSF samples. The CSF-to-plasma ratios were 4 times higher for RPV than for CAB (2.8% and 0.6%, respectively).

To further define the penetration of CAB and RPV in tissue virus reservoirs, we measured CAB and RPV concentrations in lymphoid tissue, the lower gastrointestinal (GI) tract, and the brain collected at necropsy from a SHIV-infected female macaque treated with a single intramuscular injection of Cabenuva® (also administered at 50 mg/kg CAB LA and 200 mg/kg RPV LA). Supplementary Table 1 shows drug concentrations observed 3 weeks after injection. CAB and RPV achieved high concentrations in lymphoid tissues (median of 975 ng/g and 773 ng/g, respectively) and in the GI tract (medians of 695 ng/g and 666 ng/g, respectively). Consistent with the higher CSF-to-plasma penetration ratio, the median concentration of RPV in brain tissues was 5.8 times higher than that of CAB (340 ng/g compared to 58 ng/g).

### Concentrations of dATP and dCTP in PBMCs and tissues

TFV-DP and FTC-TP compete with dATP and dCTP, respectively, as substrates for incorporation by reverse transcriptase. As a measure of the antiviral activity of FTC-TP and TFV-DP in cells, we calculated dATP/TFV-DP and dCTP/FTC-TP ratios in PBMCs and tissues. High ratios indicate a reduced likelihood of FTC-TP or TFV-DP to block reverse transcription[22]. Figure 3f shows a comparative analysis of dNTP/nucleotide ratios in PBMCs, LNMCs, and rectal tissues collected during treatment. In the case of dATP/TFV-DP, the lowest ratios were seen in LNMCs followed by PBMCs and rectal tissues ($p = 0.0044$ and $p < 0.001$ for comparisons between LNMC/PBMCs and PBMCs/rectal), suggesting that TFV-DP has the highest activity in lymphoid tissue and the least in rectal tissue. In contrast to dATP/TFV-DP, the ratio between dCTP/FTC-TP was similar in LNMCs and in PBMCs ($p = 0.367$). dCTP/FTC-TP ratios in the only two rectal biopsies with detectable FTC-TP were higher than in PBMCs or LNMCs.

### Short ART with VES also results in virus remission

We next investigated in 4 macaques if the addition of the immuno-modulator VES to a shortened eART regimen could also result in virus remission. eART with FTC/TAF/CAB LA/RPV LA was initiated after the onset of viremia, days 6–8 post-infection, and continued for 4 months. Supplementary Fig. 2 shows the adherence to oral FTC/TAF and the overall longitudinal concentrations of TFV-DP (median = 1162 (rage = 417–2093) fmols/10⁶ PBMCs) and FTC-TP (median = 553 (246–914 fmols/10⁶ PBMCs) during the entire treatment period. VES treatment was started two days after ART initiation and was given orally once a week for three months (Fig. 4a). Overall, the levels of virus replication during the first three weeks of treatment (as expressed by RNA copies/day*mL of plasma) were similar among macaques in the eART + VES and eART-only groups (median of 40 and 45, respectively,

$p = 0.686$) (Fig. 4b). The median (range) plasma RNA level at treatment initiation (day 6–8) was 3.7 (3.0–4.3) log₁₀ copies/mL with peak levels (4.2 (3.7–4.4) log₁₀ copies/mL) seen 8–12 days post-infection (Fig. 4c). Viremia was rapidly suppressed (day 19–28) and maintained below the limit of detection during the 4 months of ART and following treatment discontinuation except for one animal (MAJ) that showed a virus blip 5 months after treatment cessation (9 months post infection) (Fig. 4c). No virus rebound was observed in any of the macaques following depletion of CD8 + T cells. As noted in animals that received ART without VES, SHIV IgG responses were limited to gp140 albeit these responses were delayed compared to the eART-only group (36.5 [33–40] vs 14 [14–17] days, $p = 0.0286$) (Fig. 4d). Supplementary Fig. 3 shows the kinetics of antibody responses against gag (p27) pol (p66), env (gp140, gp130, gp41, gp36) and nef seen in the 4 macaques treated with eART + VES.

### Intact proviral DNA levels in PBMCs and tissues

Intact and defective proviral DNA levels were measured in PBMCs and tissues using the intact proviral DNA assay (IPDA)[23]. At day 21 after infection (or ~ 2 weeks after treatment initiation), the median intact proviral DNA level in PBMCs in the eART-treated macaques was 42 copies/10⁶ PBMCs (range = 15–192) compared to 9392 (4416–15,439) copies/10⁶ PBMCs in untreated animals (Table 2). Intact proviral DNA was also detected in one of the eART animals (macaque 10–47) 1 year after treatment cessation and 2 months after CD8 T cell depletion, a period that coincided with a transient viral blip in plasma (Fig. 1). In the eART+VES group, intact proviral DNA at day 21–25 after infection was only detected in one animal (macaque 42268) (Table 2).

At necropsy, none of the eART animals had detectable intact proviruses in PBMCs as opposed to the two untreated controls tested, which had 12,315 and 964 copies/10⁶ PBMCs (Table 2). Interestingly, macaque 44346 treated with eART + VES had levels of intact provirus DNA at necropsy 1 year post-infection (480 copies/10⁶ PBMCs) that were similar to untreated macaque 10–87 (964 copies/10⁶ PBMCs). However, macaque 44346 remained aviremic, while macaque 10–87 exhibited virus loads of 4.4–5.5 log₁₀ copies/mL during a period of 6 months, likely reflecting the role of RPV LA in maintaining long-term suppression.

We also measured intact proviral DNA at necropsy in lymph nodes (mesenteric, inguinal, or axillary) and spleen from the eART group, and mesenteric lymph nodes from the eART + VES group. The eART-treated animals had no detectable intact provirus in the spleen, inguinal, or axillary lymph nodes (Table 3). However, 2 of the 4 eART animals (macaques 10–47 and 28025) and 1 of the 4 eART + VES-treated animals (macaque 44346) had detectable intact provirus in mesenteric lymph nodes. As noted in PBMCs, macaque 44346 (eART + VES group) had a frequency of intact proviruses in mesenteric lymph nodes that was similar to that seen in untreated control macaque 10–87 (Table 3).

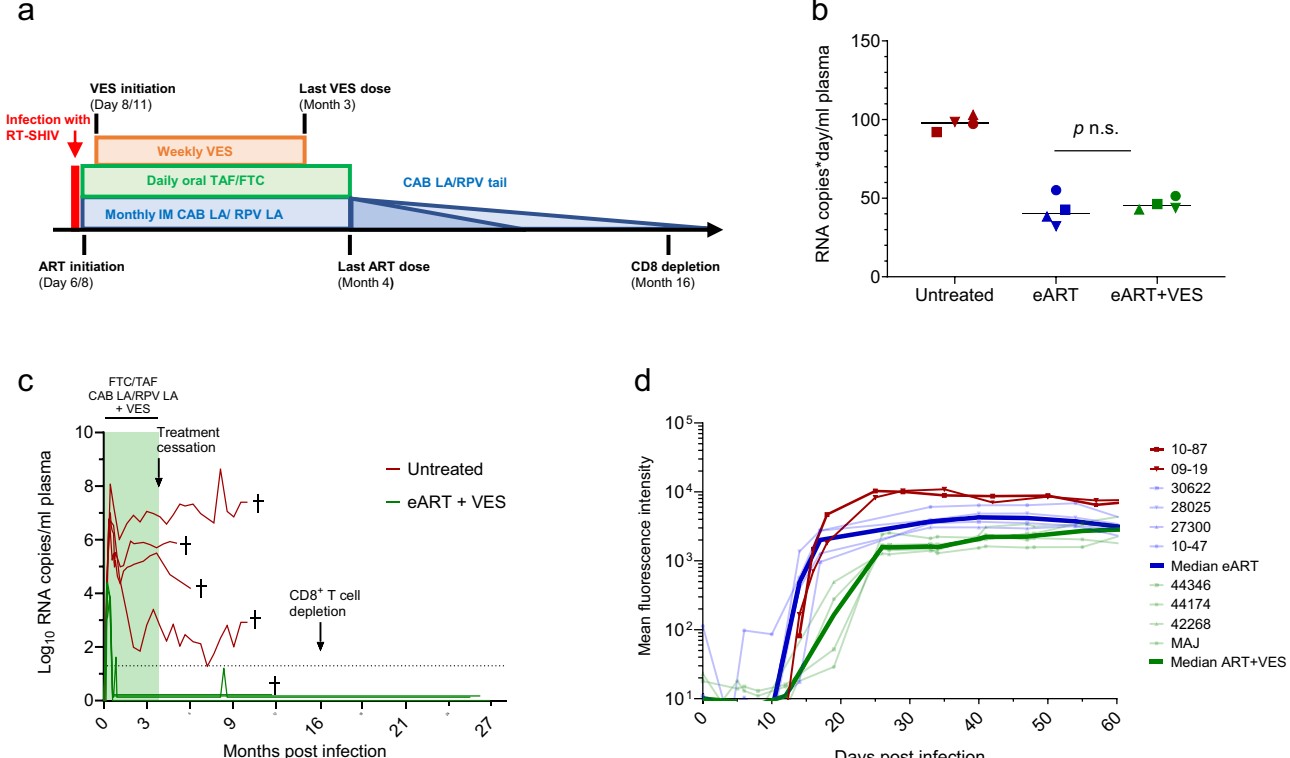

**Fig. 4 | Infection outcome after early treatment with ART and VES. a** Treatment with FTC/TAF/CAB LA/RPV LA was initiated at day 6/8 post-infection and treatment with VES at day 8/11. FTC/TAF/CAB LA/RPV LA was administered for 4 months and VES for 3 months. Transient CD8 depletion was done 16 months after treatment initiation (or 12 months after treatment discontinuation). **b** Area under the curve values for SHIV RNA ($AUC_{0-19days}$) in untreated controls (red) and macaques treated with eART only (blue) and eART + VES (green). Each data point reflects AUC values seen in one macaque. $AUC_{0-19days}$ values were similar among macaques in the

eART + VES and eART-only groups ($p = 0.686$; two-sided Wilcoxon Rank Sum test). **c** SHIV infection dynamics during treatment and after discontinuation. The four untreated controls are shown in red for comparison. **d** SHIV plasma IgG levels against gp140 during acute infection in macaques treated with eART + VES (individual animals in light green; dark green denotes median), eART alone (individual animals in light blue; dark blue denotes median), and in two untreated controls (red lines). Source data are provided as a Source Data file.

## Discussion

Conventional ART regimens are designed to durably suppress viremia in blood but do not prevent HIV or SIV rebound after treatment discontinuation even when given early after infection[8,9,24]. We reasoned that alternative ART modalities are needed to increase the likelihood of virus remission. In this study, we document long-term post-treatment viral control in macaques treated at the onset of viremia with a suppressive regimen containing oral and long-acting antiretrovirals. Remission was observed in all study animals irrespective of the length of ART (4 months or 1 year) and was maintained for about 2 years until necropsy. These data contrast findings in a similar intrarectal model of infection with SIV in which early treatment with suppressive subcutaneous FTC, TDF, and DTG was not effective in achieving remission[8,9]. We show that acute infection dynamics and chronic viremias in untreated RT-SHIV infections were similar to those seen with SIV suggesting that differences in virus pathogenicity may not explain the different outcomes. Also, two of our untreated animals progressed to simian AIDS within only 6 months. We posit that the robust post-treatment control seen in our study reflects early treatment initiation, the use of an optimized ART regimen that better penetrates tissue reservoir sites, and the addition of long-acting drug formulations that provide prolonged antiviral activity after treatment cessation. We detail the pharmacologic advantages of the regimen by conducting extensive pharmacokinetic studies. We found that the concentrations of TFV-DP in PBMCs and lymphoid tissues were about 10-fold higher than those achieved with TDF highlighting the pharmacological advantage of TAF vs TDF in blocking infection in lymphoid tissues[10]. We also show dATP/TFV-DP ratios that are indicative of efficient RT

inhibition[22]. We detect both CAB and RPV at high concentrations in lymphoid tissues and the GI tract. Importantly, we show that this regimen distributes two potent drugs, CAB and RPV, at inhibitory concentrations in CSF and brain tissues that are not provided by the conventional ART regimen with FTC, TDF, and DTG. Based on the in vitro $EC_{90}$ values for RPV and CAB of 0.66 ng/mL and 0.34 ng/mL, respectively, the RPV and CAB concentrations in CSF were about 6 times and 17 times above the $EC_{90}$, respectively. In brain tissues, RPV and CAB concentrations were ~500 and ~100 times above the in vitro $EC_{90}$, respectively, highlighting the antiviral activity of RPV and CAB in brain tissues. Therefore, our pharmacokinetic observations point to an efficient drug distribution in virus reservoir sites at a very early stage of reservoir seeding. A second critical feature of this regimen is the inclusion of long-acting drugs that prolong antiviral activity for at least two years after treatment cessation. Overall, we show the capability of this novel ART modality to achieve durable virus remission.

We also note that our ART regimen is clinically relevant and easily translatable to humans. Antiretroviral drug doses were adjusted by allometric scaling due to the smaller size and faster metabolisms of macaques to better mimic clinical doses in humans. FTC and TAF were administered orally as in humans to better model drug biodistribution in tissues, and CAB LA and RPV LA were given by intramuscular injection. The addition of the NNRTI RPV also incorporates a new class of drugs that, to our knowledge has never been part of ART regimens in macaque models of remission and cure. This was significant since NNRTI-based regimens have been associated with delayed post-treatment virus rebound in some clinical studies[25]. More recently, RPV was also found to induce clearance of latently infected cells

**Table 2 | Intact and defective proviral DNA in PBMCs during acute infection and at necropsy**

| | Macaque ID | Provirus (copies/$10^6$ PBMCs) | | | |
|---|---|---|---|---|---|
| | | Intact | 3' defective | 5' defective | Total |
| **eART** | | | | | |
| Day 21 post-infection | 10–47 | 192 | 64 | <60* | 256 |
| | 27300 | 59 | <2 | 7 | 66 |
| | 28025 | 25 | <3 | <3 | 25 |
| | 30622 | 15 | 9 | 1 | 25 |
| 1 year after treatment cessation | 10–47 | 4 | <1 | <1 | 4 |
| | 27300 | <1 | <1 | <1 | <1 |
| | 28025 | <1 | <1 | <1 | <1 |
| | 30622 | <2 | 2 | 2 | 2 |
| 2 months after CD8 + T cell depletion | 10–47 | 2 | <2 | <2 | 2 |
| | 27300 | <1 | <1 | <1 | <1 |
| | 30622 | <2 | <2 | <2 | <2 |
| Necropsy | 10–47 | <2 | 2 | <2 | 2 |
| | 27300 | <2 | <2 | <2 | <2 |
| | 28025 | <2 | <2 | 2 | 2 |
| | 30622 | <2 | <2 | 2 | 2 |
| **eART + VES** | | | | | |
| Day 21–25 post-infection | MAJ | <93 | <93 | <93 | <93 |
| | 44174 | <2 | 2 | <2 | 2 |
| | 42268 | 17 | 4 | <4 | 21 |
| | 44346 | <50 | <50 | <50 | <50 |
| Necropsy | MAJ | <10 | <10 | <10 | <10 |
| | 44174 | <3 | <3 | <3 | <3 |
| | 42268 | <2 | <2 | 7 | 7 |
| | 44346 | 480 | <1 | 43 | 523 |
| **Untreated controls** | | | | | |
| Day 21 | 09–19 | 8182 | 202 | 593 | 8977 |
| | 10–87 | 4416 | <2 | 311 | 4727 |
| | 12C189 | 15,439 | 453 | 784 | 16,676 |
| | 40526 | 10,601 | <145 | 472 | 11,072 |
| Necropsy | 09–19 | 12,315 | <2 | 481 | 12,797 |
| | 10–87 | 964 | 44 | 71 | 1078 |

*the frequency of undetectable was calculated based on assay cell input.

**Table 3 | Intact and defective proviral DNA in tissues at necropsy**

| | Macaque ID | Provirus (copies/$10^6$ cells) | | | |
|---|---|---|---|---|---|
| | | Intact | 3' defective | 5' defective | Total |
| **eART** | | | | | |
| Mesenteric LN | 10–47 | 59 | <48* | <48 | 59 |
| | 27300 | <81 | <81 | <81 | <81 |
| | 28025 | 3 | <2 | <2 | 3 |
| | 30622 | <5 | <5 | <5 | <5 |
| Inguinal LN | 10–47 | <35 | <35 | <35 | <35 |
| | 27300 | <2 | <2 | 2 | 2 |
| Axillary LN | 10–47 | <8 | 8 | <8 | 8 |
| | 28025 | <9 | <9 | <9 | <9 |
| Spleen | 28025 | <63 | <63 | 65 | 65 |
| | 30622 | <3 | <3 | 6 | 6 |
| **eART + VES** | | | | | |
| Mesenteric LN | MAJ | <4.87 | <4.87 | <4.87 | <4.87 |
| | 44174 | <2.74 | <2.74 | 6 | 6 |
| | 44346 | 1401 | 163 | 527 | 2090 |
| **Untreated controls** | | | | | |
| Mesenteric LN | 09–19 | 26,068 | 58 | 1062 | 27,183 |
| | 10–87 | 2780 | 338 | 607 | 3725 |

*the frequency of undetectable was calculated based on assay cell input.

RPV for ~ 2 years after the last dosing at subtherapeutic concentrations in our macaques should not be sufficient to prevent virus rebound under continuous residual virus replication. Therefore, the lack of virus rebound or emergence of RPV-resistant viremia in all animals despite almost two years of plasma exposure to RPV supports minimal or no residual virus replication and points to a disrupted virus reservoir, as illustrated by the 20 to >1000-fold reduction in the frequency of intact proviruses seen in treated animals within weeks of infection. These observations open the possibility for novel HIV remission approaches based on the efficient control of a disrupted reservoir using infrequent RPV LA injections (or other long-acting antiretrovirals) given perhaps every 1-2 years. Because the 4 drugs in the tested regimen are clinically available, they can be explored in clinical studies such as treatment of early or late acute infection to assess their advantage over conventional ART in inducing remission.

Our study is a treatment study of infected macaques with demonstrated viremia and, thus, is fundamentally different from prevention studies with CAB LA as PrEP in uninfected macaques in which protected animals remained seronegative and negative for SHIV RNA in plasma and SHIV DNA in PBMCs[27,30]. Interestingly however, the dynamics of early infection seen in our study share some characteristics with those observed in people who received CAB LA for preexposure prophylaxis (PrEP) and become infected despite on-time injections[31]. These breakthrough infections are also characterized by transient low-level RNA and DNA detection and diminished or delayed antibody responses[32]. It will be interesting to understand if CAB LA PrEP failures are associated with some degree of reservoir disruption and virus remission.

The addition of the TLR7 agonist VES to our ART regimen also resulted in durable viral remission that extended for almost 2 years. We selected VES for its ability to enhance innate and adaptive immune responses by activating NK cells, CD4 + and CD8 + T cells[33–36]. CD4 + cells that are transitioning from an effector to a memory phenotype are thought to be a primary source of the latent HIV reservoir[37–39]. We hypothesize that VES activation may limit the number of CD4 T cells transitioning to a memory phenotype, in turn reducing the pool of reservoir cells. In addition, VES-accelerated CD8 + T cell responses

through pyroptotic cell death, a mechanism that could potentially reduce the size of the latent HIV reservoir[26].

The use of the long-acting formulations of CAB and RPV was also important as these formulations have an extended pharmacological tail capable of maintaining therapeutic drug levels for several months after treatment discontinuation. We report here plasma CAB levels above the $4 \times$ PA-IC$_{90}$ for about 5-6 weeks after treatment cessation and RPV concentrations of about $3 \times$ PA-IC$_{90}$ for almost 2 years after the last injection, a length of time that would likely be extended in humans due to the longer CAB and RPV half-life[27,28]. In macaques, the clearance of CAB is much faster than in humans[27]. In humans, the estimated apparent terminal elimination half-life of RPV after a single intramuscular administration is 91–196 days[16]. Repeated RPV dosing results in significant drug accumulation[29] as we also noted in macaques. It is possible that the persistence of RPV may have contributed to post-treatment control by providing continuous antiviral activity through pyroptotic cell death of infected cells and suppression of any residual virus replication[26]. This scenario is different from treatment of established infections which requires bimonthly or monthly dosing of long-acting RPV and CAB to maintain virus suppression[21]. Persisting

might also provide opportunities for a more robust anti-HIV activity. In a recent study in macaques, early treatment with ARVs also favored the development of long-term CD8 + T cells with enhanced proliferative and SIV suppressive capacity[40]. In humans, VES has been studied as a latency-reversing agent and associated with decreases in intact proviral DNA and a modest effect on time to rebound[41]. In our study, we administered VES 2-3 days after ART initiation to minimize and purge reservoir formation from activated CD4 + T cells and found a reduction in intact proviruses in all four animals. Notably, none of the macaques showed virus rebound despite a period of treatment of only 4 months and subsequent depletion of CD8 + T cells 1 year later. It will be important to see if a shorter duration of ART treatment without VES would achieve a similar degree of virus remission and also to further investigate the effect of latency disruption during early infection.

Our study has a few limitations. Our findings in the rectal infection macaque model may not be generalizable to all routes of SHIV acquisition and vaginal, penile, and intravenous infections may require independent confirmation. Okoye et al. reported virus remission in animals infected with SIV intravenously and treated for almost two years with TDF, FTC, and DTG at day 4/5[42], but when treatment was initiated on day 6–12 almost all of the animals had viral rebound. Importantly, the same ART regimen initiated on day 3 post rectal virus challenge did not result in remission[8,9]. It is also not known if oral formulations of CAB and RPV lacking the long pharmacological tail might result in the same degree of post-treatment control. Additional studies will be required to assess these oral regimens and the importance of the pharmacologic drug tail. Likewise, it will be important to define if treatment initiated later in acute infection or after seroconversion would be as effective in maintaining post-treatment control as this would be the case for most people who initiate early HIV treatment. Lastly, although the degree and length of remission seen in our study are unprecedented, we do not know if remission would be maintained when RPV levels in plasma become undetectable.

Several important observations can be made from the analysis of intact proviral DNA. First, early treatment with FTC, TAF, CAB LA, and RPV LA with or without VES resulted in a significant reduction in the frequency of intact proviruses during acute infection. In contrast, Whitney et al. did not observe reductions in SIV DNA during early infection when FTC/TDF/DTG was initiated 7–10 days after infection[8,9]. These observations highlight the superiority of FTC, TAF, CAB LA and RPV LA combination in reducing the early virus reservoir. Second, despite the lack of detectable plasma viremia throughout the study, intact proviruses were detected in PBMCs and mesenteric lymph nodes from some of the treated animals 1.5–2 years after treatment cessation. Although it is not known if these proviruses would be capable of viral rebound, these findings suggest that RPV was sufficient to control the small virus reservoir established during early infection.

In summary, we show in macaques that a combination of early reservoir reduction, pharmacological drug optimization, and ultra-long-lasting antiviral activity that persists for years after treatment cessation can result in virus remission. Additional macaque studies with this ART combination are planned to investigate the window of treatment initiation (i.e., late acute infection or chronic infection) and treatment duration that will achieve virus remission. These results will inform new remission studies in humans using this ART combination and open a wider opportunity to test other combinations that provide ultra-long-lasting antiviral activity after treatment cessation[43].

## Methods

### Ethics statement

All animal procedures received prior approval from the Institutional Animal Care and Use Committee (IACUC) of the Centers for Disease Control and Prevention and were conducted in a United States Department of Agriculture (USDA)-registered, Office of Laboratory Animal Welfare (OLAW)-assured, and AAALAC International-accredited animal facility in accordance with the Guide for the Care and Use of Laboratory Animals. Euthanasia was performed and confirmed by the attending veterinarian. A total of 13 Rhesus macaques (10 males, 3 females) were used in the study. At the end of the study, animals were anesthetized with 1 mg/kg ketamine and euthanized using intravenous injection of Pentobarbital euthanasia solution (>100 mg/kg). Death was confirmed by monitoring the absence of heart rate and breathing. The procedure was performed in accordance with the CDC-Atlanta IACUC Policy on Euthanasia and Care Guidelines recommended by the American Veterinary Medical Association Guidelines on Euthanasia, 2020.

### Viruses, antiretroviral drugs, and dosing

RT-SHIV was selected for the study since it contains the $HIV_{HXBc2}$ RT that is sensitive to RPV. RT-SHIV was obtained through the NIH AIDS Reagent Program, Division of AIDS, NIAID, NIH[17–19]. The stock was maintained in liquid nitrogen in individual 1 mL aliquots until use. TAF and FTC were gifts from MS Hetero Labs Limited and Laurus Labs, respectively. CAB LA and RPV LA were obtained from ViiV Healthcare and Janssen, respectively. TAF (1.5 mg/kg) and FTC (20 mg/kg) were given orally once a day mixed with food[44]. CAB LA (50 mg/kg) and RPV LA (200 mg/kg) were given intramuscularly once a month[27,45]. The macaque used in the terminal PK study received CAB LA (50 mg/kg) and RPV LA (200 mg/kg) from Cabenuva®.

Animals were trained to take TAF, and FTC mixed with food prior to study initiation[46]. Macaques were observed for medication consumption, and adherence was documented using medication diaries. Adherence in the eART group was nearly perfect (99.5% or 352/356 doses consumed) during the first 3.5 months of study with imperfect adherence only noted in one animal (30622) who ate 50–90% of the dose during the first week. After a sudden drop in adherence at month 3.5, numerous food contingencies for FTC/TAF delivery were tested, including juices, sugar cookies, and chocolate chip cookie dough but had limited success[46]. In an attempt to maintain effective TFV-DP and FTC-TP concentrations, we administered FTC/TAF twice a week by gavage. Twice weekly dosing by gavage and sporadic adherence to the drug mixed with food maintained detectable levels of FTC-TP and TFV-DP through the remainder of the treatment period.

Animals in the eART + VES group were also monitored for daily FTC/TAF intake. Given the drop in TFV-DP and FTC-TP levels observed in the eART group during periods of suboptimal adherence, we incorporated subcutaneous FTC/TAF dosing (20 mg/kg and 0.2 mg/kg, respectively) when the oral medications were not consumed. On the days of anesthesia, FTC and TAF were always administered by oral gavage.

### Macaque study design

Rhesus macaques were evenly distributed according to MHC class I alleles with two Mamu-A*01 animals in the untreated control group (10–87 and 12C189), two Mamu-A*01 animals in the eART group (10–47 and 28025), and two Mamu-A*01 animals in the eART + VES group (44174 and 42268). All animals were challenged with 1 mL of RT-SHIV ($10^{3.3}$ $TCID_{50}$) intrarectally. Four untreated macaques (3 males and 1 female; median weight = 13.7 kg (range = 11.5–15.4)) were used as controls. Treatment with TAF, FTC, CAB LA, and RPV LA in the eART group (4 males; median weight = 14.5 kg (range = 11.3–15.5)) was initiated after infection was confirmed by the detection of SHIV RNA in plasma, which occurred on day 5–8 post-inoculation. Macaques in the eART group received TAF/FTC/CAB LA/RPV LA for 6 months followed by a maintenance regimen with CAB LA and RPV L/A for an additional 6 months. Macaques in the eART + VES group (3 males and 1 female; median weight = 10.8 kg (range = 5.9–12.6)) received 4 months of FTC/TAF/CAB LA/RPV LA therapy initiated 6–8 days post-infection. VES treatment was added 2 days after ART initiation and was given orally once a week (0.15 mg/kg) for three months. All treated animals were

monitored for virus rebound during 1.5 to 2 years after treatment cessation including a period of transient in vivo CD8 depletion at month 30 (eART group) or 16 (eART + VES group) with anti-CD8 antibody MT807R1 (one subcutaneous 10 mg/kg dose followed by three subsequent intravenous 5 mg/kg doses 3, 7, and 10 days later). The rhesus IgG1 recombinant anti-CD8 alpha [MT807R1] monoclonal antibody was engineered and produced by the Nonhuman Primate Reagent Resource (NIH Nonhuman Primate Reagent Resource Cat# PR-0817, RRID:AB_2716320). SHIV testing was done every 1-2 weeks to monitor for transient blips.

### SHIV RNA assay
Plasma viral load was determined by an RT-PCR assay modified to increase sensitivity to 12.5 copies/mL by using a larger volume of plasma[47]. Briefly, 2.5 mL of macaque plasma was ultracentrifuged for 45 min at 43,000 rpm at 4 °C, and viral RNA was extracted using Qiagen Viral Mini Kit per the manufacturer's protocol. Quantitative RT-PCR was performed to amplify a conserved region of the *gag* using a SuperScript III One-step RT-PCR kit.

### Intact provirus DNA assay (IPDA)
Accelevir performed an SIV-adapted IPDA® to characterize and quantify persistent RT-SHIV proviruses. Samples with a cell input below ~$10^4$ cells were excluded from the analysis. Overall, genomic DNA was isolated from a median of $4.5 \times 10^5$ lymph nodes or peripheral blood mononuclear cells (range = $9.8 \times 10^3$, $9.2 \times 10^5$). Briefly, RNA-free genomic DNA was isolated using the QIAamp DNA Mini Kit (QIAGEN). DNA concentrations were determined by fluorometry (Qubit dsDNA BR Assay Kit, Thermo Fisher Scientific), and DNA quality was determined by ultraviolet-visible spectrophotometry (QIAxpert, Qiagen). Genomic DNA was then analyzed by the SIV-adapted IPDA®[23]. Operators performed testing under Accelevir standard operating procedures and were blinded to sample and study identifiers.

### Analysis of intracellular FTC-TP, TFV-DP, and dNTPs
PBMCs were isolated from blood collected in BD Vacutainer® CPT™ Sodium Citrate Cell Preparation Tubes, and red blood cells were lysed with BioLegend RBC lysis buffer. Cell counts and viability were determined using a Guava Cell Counter and Cytosoft data acquisition and analysis software (version 6.02; Millipore Billerica, MA) or a Countess (Invitrogen). Three million viable cells were pelleted, washed with 1 mL of physiological saline, and resuspended in 500 µl ice-cold 80% methanol. Samples were vortexed for 1 min and immediately frozen at −70 °C. Prior to analysis, cellular debris was pelleted, and supernatants were transferred to new tubes. Supernatants were dried down and reconstituted in 100 µl of 50 mM ammonium acetate buffer (pH 7.0), then centrifuged at 17,000 × g to remove insoluble particulates. Intracellular dATP, dCTP, TFV-DP, and FTC-TP concentrations were measured using an automated on-line weak anion-exchange (WAX) solid-phase extraction (SPE) method coupled with ion-pair (IP) chromatography-tandem mass spectrometry (MS-MS)[48]. TFV-DP and FTC-TP were monitored through 448 → 176, 488 → 230, 495 → 136, and 468 → 112 *m/z* fragments, respectively, with [$^{13}C_5$] adenine-labeled internal standards for each analyte. Calibration curves were generated from serially diluted standards over the range from 0.25 to 10 nM. The lower limit of quantification is 10 ng/mL for TFV-DP and 25 ng/mL for FTC-TP. All calibration curves had $r^2$ values greater than 0.99[48]. In tissue homogenates, the limit of detection was 100 and 500 fmol/sample for TFV-DP and FTC-TP, respectively.

### Analysis of FTC, TFV, CAB, and RPV in plasma and tissues
The concentrations of CAB in plasma and the concentrations of FTC and TFV in tissue homogenates were measured by HPLC-MS/MS. A cocktail of methanol containing internal standards was added to precipitate proteins. After a brief centrifugation and removal of protein precipitates, the supernatant was evaporated to near dryness and re-suspended with mobile phase A. An aqueous-acetonitrile mobile-phase gradient was used to elute the drugs from a C18 column (100*1 mm, Imtakt) and into the analyzer. Drug concentrations were calculated from a standard curve with a range of 0.5–2000 ng/mL using Analyst software. The LOQ for CAB in plasma is 10 ng/mL, and the limit of quantification (LOQ) for tissue FTC and TFV is 1 ng/sample. Plasma concentrations of CAB were measured weekly. Plasma concentrations of RPV were measured 4 (9 observations), 2 (2 observations), or 3 (2 observations) weeks after the previous injection. The concentrations of RPV in plasma, CSF, and tissues and the concentrations of CAB in CSF were measured at the Clinical Pharmacology and Analytical Chemistry (CPAC) Laboratory, University of North Carolina Center for AIDS Research. The LOQ for RPV in plasma is 1 ng/mL.

### Cell surface staining and flow cytometry
PBMCs were stained for viability using the Zombie Yellow™ Fixable Viability Kit (Biolegend®) prior to cell staining with fluorochrome-conjugated antibodies as previously described[49]. Stained samples were run on an LSRII flow cytometer, and data was acquired using FACS DIVA software (BD Biosciences) and analyzed using FlowJo software (TreeStar, Inc.). The fluorochrome-conjugated antibodies are described in Supplementary Table 2.

### Luminex assay for antibody responses
The Luminex assay was performed for the detection of IgG reactivity against MagPlex magnetic COOH microspheres (Luminex, Austin, TX) coupled with the following recombinant proteins: SIV$_{mac239}$ gp140, SIV$_{mac251}$ gp130, SIV$_{mac251}$ gp41, SIV$_{mac251}$ nef, SIV$_{mac251}$ p27, HIV-1 p66, and HIV-2 gp36 (Immunodiagnostics, Inc., Woburn, MA) (Immune Technology, New York, NY). For sample addition and non-specific binding controls, coupling reactions were also performed using goat anti-human IgG (Invitrogen, Carlsbad, CA) and BSA (Sigma-Aldrich, St. Louis, MO), respectively. The coupling of proteins to the magnetic microspheres was performed as previously described[50]. All samples were tested in duplicate. A 1:50 dilution was performed on all plasma samples in preincubation buffer (PBS with 1% BSA, 0.5% poly-vinylalcohol (PVA, Sigma-Aldrich), and 0.8% polyvinylpyrrolidone (PVP, Sigma-Aldrich)) as described[51]. Diluted samples were incubated for 15 minutes at room temperature on a titer plate shaker at 600–800 rpm. A working microsphere mixture was prepared by combining $1 \times 10^5$ microspheres/mL of each coupled microsphere set in assay buffer (PBS with 1% BSA). The microsphere mixture and pre-diluted plasma samples were combined in duplicate wells of a flat-bottom 96-well plate (Thermo Fisher Scientific, Waltham, MA) at a volume of 50 µL of each/well. The plate was incubated for 30 minutes on a shaker at 600–800 rpm with protection from light. The microspheres were washed 3 times with assay buffer using a BioTek ELx50 microplate strip washer with magnet (BioTek, Winooski, VT), resuspended in 100 µL of a 1:1000 dilution of phycoerythrin (PE)-labeled, goat anti-human IgG (Sigma Aldrich), vortexed for 30 s on a plate vortexer, followed by a 30 min incubation on a shaker at 600–800 rpm with protection from light. The microspheres were washed twice with assay buffer, then resuspended in 125 µL of assay buffer and analyzed on the Luminex 200 System (Luminex, Austin, TX). Median fluorescent intensity (MFI) values were generated for each sample and duplicate samples MFIs were averaged.

Positive control was created by pooling plasma from six SHIV-positive rhesus macaques and was included on each assay plate. The positive control was positive for all SIV/SHIV-specific analytes included in the testing panel. Cutoff values were calculated for each analyte using SHIV negative rhesus macaque plasma samples. The cutoff value for each analyte was determined using the following formula: average MFI value of the negative samples + 2 standard deviations.

## Statistical analysis

Two-sided Wilcoxon Rank Sum tests were used to compare the medians between the two groups. We used the exact test for each of the comparisons and assumed that our data followed a non-normal curve due to small sample sizes. Medians and ranges (versus means and square roots) were compared for each group due to this assumption. Furthermore, due to the small sample sizes, we decided not to include any covariates in our comparisons. Data collection was performed using Microsoft Excel and imported into SAS software version 9.4 for analysis.

## Reporting summary

Further information on research design is available in the Nature Portfolio Reporting Summary linked to this article.

# Data availability

All data supporting the findings of this study are available within the paper and its Supplementary Information. Any additional requests for information can be directed to the corresponding author. The antibodies used for flow cytometry analysis are provided in Supplementary Table 2. Source data are provided in this paper.

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

## Acknowledgements

We thank Mara Sterling and Kenji Nishiura for their help in processing some of the macaque specimens, James L. Weed for training macaques to ingest FTC and TAF orally, Patrick Mills, and Rex Howard for performing some of the veterinary procedures, and Shanon Bachman, Frank Deyounks, Ryan Johnson, and Kristen Kelley for performing animal procedures. We also thank Hetero Drugs Ltd., and Laurus Labs Ltd for providing FTC and TAF. The following reagent was obtained through the NIH HIV Reagent Program, Division of AIDS, NIAID, NIH: RT-Simian-Human Immunodeficiency Virus, ARP-11342, contributed by Dr. Thomas North and Dr. Joseph Sodroski. The Anti-CD8 alpha [MT807R1] antibody used in this study was provided by the NIH Nonhuman Primate Reagent Resource (P40 OD028116). The findings and conclusions of this manuscript are those of the authors and do not necessarily represent the official views of the CDC. This work was funded with CDC intramural funds (to J.G.G.-L.). P.W. is a former Janssen employee (retired).

## Author contributions

M.B.D., P.W., W.S., W.H., and J.G.G.-L. were responsible for project conceptualization. M.B.D., C.D., A.H., D.R., S.R., A.S.K., S.S., J.M., J.C., D.K., K.C., and J.G.G.-L. performed the investigations. Formal analysis was done by M.B.D., A.S.K., G.K., Y.P., and J.G.G.-L. Data curation was done by M.B.D. and A.S.K. Funding acquisition and supervision by J.G.G.-L. and project administration by MBD and J.G.G.-L. The writing was done by M.B.D., W.H., and J.G.G.-L.

## Competing interests

J.G.G.-L., M.B.D., W.H., P.W., and W.S. are named in a US Patent Application (No. 18/002,494) entitled "Methods for achieving viral remission using long-acting antiretroviral agents". The remaining authors declare no competing interests.

## Additional information

**Article** https://doi.org/10.1038/s41467-024-54783-0

