## [Peer Review file · Nature Communications]

SHIV remission in macaques with early treatment initiation and ultra long-lasting antiviral activity

Corresponding Author: Dr J. Gerardo Garcia-Lerma

Version 0:

Reviewer comments:

Reviewer #1

(Remarks to the Author)

The manuscript by Daly et al studied the effects of early treatment (5 to 6 days pi) of SHIV-infected rhesus macaques with a combination of 4 drugs (daily oral FTC/TAF and monthly IM long-acting regimen cabotegravir and rilpivirine). Remission was observed after animals were treated for up to 12 months and followed up for up to 2 years with treatment cessation, a CD8+ T cell depletion was conducted at month 30 pi. Animals were euthanized at month 34 pi. Additionally, eART+VES experiment was conducted with shorter ART. The authors monitored plasma viral loads throughout of the course of the experiment, examined drug concentrations in multiple tissues and found CAB LA and RPV LA in the brain with relatively high concentration. Viral reservoirs were measured by intact proviral DNA levels in PBMCs and some tissues, SHIV antibodies were evaluated, but cellular immune responses were lacking. Overall, this is a very interesting study with the use of long-acting CAB LA and RPV LA in the RT-SHIV NHP model. The remission period was prolonged by this 4-drug combination treatment. However, the very early treatment starting at 5 /6 days post infection may not be feasible and translational in humans although this is a proof-of-concept that the earlier treatment the better. It would be interesting to continue to follow up until the rebound appeared (at least another 2-3 months, as it is unlikely this is a sterilized cure). The study had very small sample size which lacks statistical power and in general the study was observational without clear underlying mechanisms.

My comments are listed below:

In Table 2, intact and defective proviral DNA was measured with the unit of 1×10^6 cells, presumably these cells were not purified CD4+ T cells, which may not accurately reflect the levels between groups as CD4+ numbers and percentages could be varied with or without different intervention between groups.

The same issue for Table 3, and there is no unit was provided. For eART+VES group, it's unclear why no other lymphoid tissues or even more tissues/compartments (i.e., the brain) were examined. While mesenteric LN was very important to test, lack of data on other tissues may not reflect the viral reservoirs in the eART+VES experimental group.

In Figure 4D, it's hard to see how many days of difference or delay between eART+VES and eART-only.

Figure 5 could be moved to supplementary materials as there is no positive observation and only one sentence described the figure.

Although animals were followed up for up to 32 months post infection, no results on side effects and/or toxicity of the combination treatment (especially CAB LA and RPV LA) were considered and provided.

Reviewer #2

(Remarks to the Author)

1. Introduction and elsewhere (e.g., Discussion, 3rd paragraph). The authors refer to "terminal elimination" half-lives for CAB and RPV. However, the terminal portion of the concentration–time profile for these long-acting drugs represents ongoing

absorption (not elimination as would be typical) from the injection site; this is known as flip-flop pharmacokinetics. See, for example, Spreen W. *J Acquir Immune Defic Syndr* 2014;67:481–486. It would be more accurate to describe these half-lives as “apparent elimination” or “apparent terminal elimination”.

2. Introduction, last paragraph. The sentence “Animals were treated after onset of viremia ” should be prefaced with “One group of”.

3. Methods.

3.1. The authors refer to a period of suboptimal adherence to oral TAF/FTC occurring after the first 3.5 months of therapy in the ART-only animals. Suboptimal adherence also seems to have occurred in the ART/VES animals. Describe how the suboptimal adherence was determined/measured; state the number of animals in both groups that had suboptimal adherence and how long this suboptimal adherence persisted.

3.2. Further, the authors used 2 different approaches to manage suboptimal adherence. Please explain why given that the 2 approaches are very different (twice weekly oral gavage vs. daily SC injection).

3.3. The authors provide no information on the effect of suboptimal adherence on the Results. PBMCs were obtained during the study and analyzed for TFV-DP and FTC-TP. The timing of when they were obtained in relation to dosing and throughout the study needs to be given. These levels would provide some information on the affect of non-adherence. Figure 3D shows the TFV-DP and FTC-TP PBMC concentrations, and the variability is quite pronounced, and may well be the consequence of suboptimal adherence. These could be stratified by TAF/FTC dosing and adherence (daily oral and twice weekly gavage).

3.4. The PBMC concentrations shown in Fig 3 are apparently only for the ART-only group. Were PBMC concentrations obtained in the ART/VES group? If so, they should be presented as well. If not, then so state.

3.5. Tissue concentrations were apparently quantified from tissue homogenates, but this should be specifically stated.

4. Results.

4.1. The authors state the range of TFV-DP concentrations were within the range observed in humans. However, the data cited for this range (reference #12) is for peak concentrations and not trough (median 523 fmol/10⁶ cells for the 25mg TAF dose) which would better reflect the range. Unless all the PBMC samples collected in the macaques were at peak this statement on comparability should be revised.

4.2. The authors need to provide a human comparison for the PBMC concentrations of FTC-TP (as they have done for TFV-DP), which were much lower in these macaques than in humans. Using their reference #12, the median FTC-TP concentrations were 5149 and 2060 fmol/10⁶ cells for peak and trough, respectively.

4.3. Table 1 gives the concentrations of CAB and RPV in CSF. The corresponding plasma concentrations need to be provided as well to support the calculation of CSF-to-plasma ratios and inform the reader of plasma concentrations 4-weeks after last injection. The section in Results beginning “Based on the in vitro EC90” through the end of the paragraph would be more appropriate in the Discussion. This will allow the results of the tissue concentrations to follow, and in particular the brain tissue concentrations, which are relevant to the CSF concentrations.

5. Discussion.

5.1. In paragraph 1, the authors should mention their findings on the CSF concentrations of CAB and RPV in addition to the brain tissue concentrations, and comment on CSF concentrations relative to published human data, see Letendre, doi: 10.1093/jac/dkz504 .

5.2. The authors have referenced work supporting CSF concentrations as a surrogate for brain concentrations in NHPs. Above in 4.3, I've suggested they move that section from Results to Discussion. First, I think the authors should comment on the agreement of their own CSF and brain tissue data for CAB and RPV. I'm uncertain of the value of the comments that were originally in Results relative to the Nagaya publication (reference #23) given their own data with measured CSF and brain tissue values. If mention of reference #23 is retained, they need to discuss the strength of the agreement found, importantly a general overestimation from CSF concentrations up to almost 3-fold higher. Clinically, this is problematic as, for example, a 2-fold higher or lower brain tissue concentration may have a quite different therapeutic effect. Some other relevant animal and human work that is also informative are: Srinivas, doi: 10.1080/00498254.2018.1539278 ,and Ferrara, doi: 10.1097/QAD.0000000000002628 .

5.3. In paragraph 1, the authors state “long-acting drugs that prolong antiviral activity”. To this reviewer, this sentence reads as attributing the prolonged effect (no virus rebound) to the long-acting drugs. But CAB and RPV did not both provide “therapeutic” plasma concentrations for two years after treatment discontinuation. Figure 3 clearly shows CAB concentrations less than putative plasma target by 5-ish weeks after treatment discontinuation. The earlier sentence “We posit that the robust post-treatment control” seems the more accurate potential explanation.

5.4. The authors might include a mention that the clearance of CAB in NHPs is much faster than in humans.

6. Reference 22. I show this reference was published in December 2020 not 2021. Please check.

7. Figure 2. May be useful to give include identifiers for the 2 control macaques in the legend. Could also define eART (early ART) in the legend where the ART regimen is given.

8. Figure 3. In the legend, include the time (usual) that the samples were obtained relative to dosing (e.g., predose) for panels A, B, C and D.

9. Suppl Table 1. I'm unsure about the appropriateness of calculating median values of CAB and RPV concentrations for the tissue groupings (eg, the various lymph nodes) in this table. Doing so seems to presume some expectation of comparability when it is known there are regional differences among these tissues with regard to drug penetration.

Reviewer #3

(Remarks to the Author)

Daly et al. report on the efficacy of early treatment in SHIV-infected macaques using an ART combination that includes two long-acting antiretrovirals and the immune-stimulating agent VES (the latter in only one of the study arms). The post-treatment control of viral loads displayed by the treated macaques is quite impressive, suggesting that this drug combination

could potentially induce a drastic remission, if not a cure, when adopted during the acute phase of infection.

The main drawback of the study is the lack of sufficient controls to determine the contribution of each treatment component to the observed remission, particularly since one of the drugs administered persisted in plasma throughout the entire post-therapy follow-up. Indeed, viral load control in macaques has recently been described upon treatment during acute infection without long-acting antiretrovirals (<https://pubmed.ncbi.nlm.nih.gov/38212337/>), although the mechanism of this control appears different from that described in the current study. Moreover, it is difficult to dissect the potential effects of the immunomodulatory treatment with VES, given that the duration of ART treatment was also altered in parallel to the addition of this compound.

Overall, despite these limitations, the study remains relevant and interesting due to the significant potential implications of the results obtained. Addressing the following points could help clarify the interpretation of some of the evidence presented:

1. Can any data be provided on rilpivirine persistence in tissues during the post-therapy follow-up?
2. Do the treated animals that displayed intact proviral DNA overlap with those displaying viral blips during follow-up or CD8 T cell depletion?
3. It would be important to determine whether viral expression can be reactivated in vitro in the samples with detectable intact proviral DNA (e.g., using a viral outgrowth assay) after a washing period to reduce RPV concentration and in the presence of a strong reactivating stimulus (e.g., PMA/Ionomycin).
4. The contribution of VES to the observed effects is unclear. While adding control macaques could prove difficult, it would be relatively straightforward to test whether VES treatment was indeed associated with increased T-cell-mediated immune responses (e.g., through an ELISpot assay) or NK cell responses.

Reviewer #4

(Remarks to the Author)

I co-reviewed this manuscript with one of the reviewers who provided the listed reports. This is part of the Nature Communications initiative to facilitate training in peer review and to provide appropriate recognition for Early Career Researchers who co-review manuscripts

Version 1:

Reviewer comments:

Reviewer #1

(Remarks to the Author)

Although some requests might not be able to met due to unavailable samples or due to the size of animal numbers, the manuscript has been greatly improved. The authors have responded to all the reviewers critiques and made great changes accordingly. No further request from this reviewer.

Reviewer #2

(Remarks to the Author)

The authors have responded substantively to the comments of the reviewers. I have no further comments or recommendations.

Reviewer #3

(Remarks to the Author)

In light of the lack of materials and cells acknowledged by the Authors, I have no further questions.

Reviewer #4

(Remarks to the Author)

REVIEWER COMMENTS

Reviewer #1 (Remarks to the Author):

The manuscript by Daly et al studied the effects of early treatment (5 to 6 days pi) of SHIV-infected rhesus macaques with a combination of 4 drugs (daily oral FTC/TAF and monthly IM long-acting regimen cabotegravir and rilpivirine). Remission was observed after animals were treated for up to 12 months and followed up for up to 2 years with treatment cessation, a CD8+ T cell depletion was conducted at month 30 pi. Animals were euthanized at month 34 pi. Additionally, eART+VES experiment was conducted with shorter ART. The authors monitored plasma viral loads throughout of the course of the experiment, examined drug concentrations in multiple tissues and found CAB LA and RPV LA in the brain with relatively high concentration. Viral reservoirs were measured by intact proviral DNA levels in PBMCs and some tissues, SHIV antibodies were evaluated, but cellular immune responses were lacking. Overall, this is a very interesting study with the use of long-acting CAB LA and RPV LA in the RT-SHIV NHP model. The remission period was prolonged by this 4-drug combination treatment. However, the very early treatment starting at 5 /6 days post infection may not be feasible and translational in humans although this is a proof-of-concept that the earlier treatment the better. It would be interesting to continue to follow up until the rebound appeared (at least another 2-3 months, as it is unlikely this is a sterilized cure). The study had very small sample size which lacks statistical power and in general the study was observational without clear underlying mechanisms.

We thank the reviewer for the thoughtful comments and appreciate that he/she understands that this is a proof-of-concept observational study.

My comments are listed below:

In Table 2, intact and defective proviral DNA was measured with the unit of 1×10^6 cells, presumably these cells were not purified CD4+ T cells, which may not accurately reflect the levels between groups as CD4+ numbers and percentages could be varied with or without different intervention between groups.

The reviewer is correct, and we did not use purified CD4+ T cells for this analysis given the small blood volumes collected during the study. To minimize confusion, we have changed the units in Table 2 to copies/ 10^6 PBMCs. We have also added new IPDA data on the 2 missing eART+VES animals at baseline (42268 and 44346) and the one missing eART+VES animal at necropsy (42268). These new data help to better appreciate the levels of intact proviral DNA in each group and further highlights the low levels seen in treated animals compared to the untreated group. Please note that we made a minor editorial change and moved the column of intact proviruses next to the animal ID.

The same issue for Table 3, and there is no unit was provided. For eART+VES group, it's unclear why no other lymphoid tissues or even more tissues/compartments (i.e., the brain) were

examined. While mesenteric LN was very important to test, lack of data on other tissues may not reflect the viral reservoirs in the eART+VES experimental group.

We have now added the units to Table 3 which in this case are copies/10⁶ cells. We agree that adding other tissue compartments in the eART+VES would have provided a better understanding of tissue reservoirs in this group. Unfortunately, we don't have these data. We indicate in the text (page 9; last paragraph) that the analysis of intact proviruses in tissues from the eART+VES group was limited to mesenteric lymph nodes.

In Figure 4D, it's hard to see how many days of difference or delay between eART+VES and eART-only.

We have modified the x-axis of Figure 4D to show only the first 2 months of treatment. We hope this change helps to better visualize the delay in gp140 responses seen in the eART+VES group. Please note that the entire kinetics of gp140 responses in the eART+VES animals are still shown in the new Supplementary Figure 3.

Figure 5 could be moved to supplementary materials as there is no positive observation and only one sentence described the figure.

As requested, Figure 5 has been moved to supplementary materials (Supplementary Figure 3).

Although animals were followed up for up to 32 months post infection, no results on side effects and/or toxicity of the combination treatment (especially CAB LA and RPV LA) were considered and provided.

We did not perform a detailed analysis of side effects or drug toxicities since these four ARV drugs have been extensively studied in macaques and humans and are generally safe and well tolerated. However, we did not observe any significant weight loss over the course of treatment (Daly et al, PLoS One 2019). In addition, our facility veterinarians did not flag any apparent toxicities during biannual physicals that included complete CBC and blood biochemistry. Although reassuring, we cannot draw any conclusion given the small sample size. We feel that the safety of this 4-drug combination analysis would be best addressed in humans.

Reviewer #2 (Remarks to the Author):

We thank the reviewer for all these excellent comments which have helped improve the manuscript.

1. Introduction and elsewhere (e.g., Discussion, 3rd paragraph). The authors refer to “terminal elimination” half-lives for CAB and RPV. However, the terminal portion of the concentration–time profile for these long-acting drugs represents ongoing absorption (not elimination as would be typical) from the injection site; this is known as flip-flop pharmacokinetics. See, for example, Spreen W. J Acquir Immune Defic Syndr 2014;67:481–486. It would be more accurate to describe these half-lives as “apparent elimination” or “apparent terminal elimination”.

We thank the reviewer for this note and have modified the text accordingly (Page 3; last sentence) and page 11; second paragraph).

2. Introduction, last paragraph. The sentence “Animals were treated after onset of viremia ” should be prefaced with “One group of”.

We clarify that both the eART and the eART+VES animals were treated after the onset of viremia. To minimize confusion, we now specify in Methods and Results that the eART+VES group initiated treatment at days 6/8 post-infection (page 8 and page 16). We further highlight this in the last paragraph of the introduction.

3. Methods.

3.1. The authors refer to a period of suboptimal adherence to oral TAF/FTC occurring after the first 3.5 months of therapy in the ART-only animals. Suboptimal adherence also seems to have occurred in the ART/VES animals. Describe how the suboptimal adherence was determined/measured; state the number of animals in both groups that had suboptimal adherence and how long this suboptimal adherence persisted.

Adherence was determined by direct observation. Macaques were observed for medication consumption and adherence was documented using medication diaries. The adherence diaries of the eART animals have been reported in the manuscript by Daly et al (reference 47). We now add on pages 15-16 a short description of the strategy use to maintain effective TFV-DP and FTC-TP concentrations during periods of suboptimal adherence. Overall, adherence was nearly perfect (99.5% or 352/356 doses consumed) during the first ~3.5 months, with imperfect adherence only noted in one animal (30622) who ate 50–90% of the dose during the first week of treatment. Adherence suddenly dropped after ~3.5 months of treatment and followed the patterns described in reference 47. The adherence diaries in the animals treated with eART+VES have not been previously reported and are now described in a new Supplementary Figure 2.

3.2. Further, the authors used 2 different approaches to manage suboptimal adherence. Please explain why given that the 2 approaches are very different (twice weekly oral gavage vs. daily SC injection).

The reviewer is correct in that we used two different approaches to manage suboptimal adherence. In the eART group we attempted to regain compliance by utilizing preplanned food contingencies to deliver FTC/TAF, including juices and cookie dough but had sporadic success. In an attempt to maintain effective TFV-DP and FTC-TP concentrations in these animals, we administered FTC/TAF twice a week by gavage. Twice weekly dosing by gavage and sporadic adherence to drug mixed with food maintained detectable levels of FTC-TP and TFV-DP through the remainder of the study, although levels were lower than during periods of full adherence. This is now described in page 15-16 of the methods.

Since the eART and the eART+VES groups were not done concurrently, we used the lessons learned in the eART group to incorporate subcutaneous dosing as a better option to manage suboptimal adherence in the eART+VES group. This approach was sufficient to maintain appropriate TFV-DP and FTC-TP levels when suboptimal adherence was observed as shown in

the new Supplementary Figure 2. We have expanded the description of these approaches in the Methods (pages 15 and 16) to better inform the reader.

3.3. The authors provide no information on the effect of suboptimal adherence on the Results. PBMCs were obtained during the study and analyzed for TFV-DP and FTC-TP. The timing of when they were obtained in relation to dosing and throughout the study needs to be given. These levels would provide some information on the affect of non-adherence. Figure 3D shows the TFV-DP and FTC-TP PBMC concentrations, and the variability is quite pronounced, and may well be the consequence of suboptimal adherence. These could be stratified by TAF/FTC dosing and adherence (daily oral and twice weekly gavage).

We agree and have now stratified the description of TFV-DP and FTC-TP levels according to periods of adherence. We include this new analysis in Figure 3 (panels D and E) and describe the data in the results (page 7; first paragraph).

3.4. The PBMC concentrations shown in Fig 3 are apparently only for the ART-only group. Were PBMC concentrations obtained in the ART/VES group? If so, they should be presented as well. If not, then so state.

Yes, FTC-TP and TFV-DP levels were also measured in ART+VES group animals. The results have been added to Supplementary Figure 2 that also shows that overall TFV-DP and FTC-TP levels were similar between the eART and the eART + VES group.

3.5. Tissue concentrations were apparently quantified from tissue homogenates, but this should be specifically stated.

We now indicate this on page 18 of the methods.

4. Results.

4.1. The authors state the range of TFV-DP concentrations were within the range observed in humans. However, the data cited for this range (reference #12) is for peak concentrations and not trough (median 523 fmol/10⁶ cells for the 25mg TAF dose) which would better reflect the range. Unless all the PBMC samples collected in the macaques were at peak this statement on comparability should be revised.

We thank the reviewer for this observation. Our TFV-DP data are trough concentrations. We have revised this paragraph accordingly to better describe the results (page 7, first paragraph).

4.2. The authors need to provide a human comparison for the PBMC concentrations of FTC-TP (as they have done for TFV-DP), which were much lower in these macaques than in humans. Using their reference #12, the median FTC-TP concentrations were 5149 and 2060 fmol/10⁶ cells for peak and trough, respectively.

The FTC-TP values are also trough concentrations, and this paragraph has been revised (page 7, first paragraph).

4.3. Table 1 gives the concentrations of CAB and RPV in CSF. The corresponding plasma concentrations need to be provided as well to support the calculation of CSF-to-plasma ratios and inform the reader of plasma concentrations 4-weeks after last injection. The section in Results beginning “Based on the in vitro EC90” through the end of the paragraph would be more appropriate in the Discussion. This will allow the results of the tissue concentrations to follow, and in particular the brain tissue concentrations, which are relevant to the CSF concentrations.

As requested, we have now added plasma concentrations of CAB and RPV to Table 1. We have also moved the last section of this paragraph into the discussion (page 10).

5. Discussion.

5.1. In paragraph 1, the authors should mention their findings on the CSF concentrations of CAB and RPV in addition to the brain tissue concentrations, and comment on CSF concentrations relative to published human data, see Letendre, doi: 10.1093/jac/dkz504 .

We would prefer not to add this comparison since macaque CSF RPV and CAB values were obtained 4 weeks after dosing (~4 and 6 ng/ml) and the human data in the Letendre paper shows levels at 7-10 days after injection (~2 and 10 ng/ml).

5.2. The authors have referenced work supporting CSF concentrations as a surrogate for brain concentrations in NHPs. Above in 4.3, I’ve suggested they move that section from Results to Discussion. First, I think the authors should comment on the agreement of their own CSF and brain tissue data for CAB and RPV. I’m uncertain of the value of the comments that were originally in Results relative to the Nagaya publication (reference #23) given their own data with measured CSF and brain tissue values. If mention of reference #23 is retained, they need to discuss the strength of the agreement found, importantly a general overestimation from CSF concentrations up to almost 3-fold higher. Clinically, this is problematic as, for example, a 2-fold higher or lower brain tissue concentration may have a quite different therapeutic effect. Some other relevant animal and human work that is also informative are: Srinivas, doi: 10.1080/00498254.2018.1539278 ,and Ferrara, doi: 10.1097/QAD.0000000000002628 .

As noted in our response to comment 4.3, we moved this part of the results to the discussion section. In response to this comment, we removed the reference to the Nagaya publication and specify how the CSF and brain tissue concentrations relate to the vitro EC90 of RPV and CAB (page 10).

5.3. In paragraph 1, the authors state “long-acting drugs that prolong antiviral activity”. To this reviewer, this sentence reads as attributing the prolonged effect (no virus rebound) to the long-acting drugs. But CAB and RPV did not both provide “therapeutic” plasma concentrations for two years after treatment discontinuation. Figure 3 clearly shows CAB concentrations less than putative plasma target by 5-ish weeks after treatment discontinuation. The earlier sentence “We posit that the robust post-treatment control” seems the more accurate potential explanation.

We have now added the words “early treatment initiation” to re-emphasize that the results might be explained by a combination of all these three factors (early treatment, regimen optimization, and long-lasting ARVs). We agree in that CAB levels decline quickly compared to RPV levels and

may not have played the same role in post-treatment control. However, we feel that RPV at subtherapeutic concentrations may have contributed by providing continuous antiviral activity through pyroptotic cell death of infected cells or suppression of any residual virus replication as we mention on page 11 of the discussion.

5.4. The authors might include a mention that the clearance of CAB in NHPs is much faster than in humans.

We now mention this in the discussion (page 11) and provide a reference (ref 27).

6. Reference 22. I show this reference was published in December 2020 not 2021. Please check.

We reviewed the record of this publication, and it indicates that this work was published in 2021 although the epub was on Dec 9, 2020. The full reference in PubMed is: Overton ET, Richmond G, Rizzardini G, et al. Long-acting cabotegravir and rilpivirine dosed every 2 months in adults with HIV-1 infection (ATLAS-2M), 48-week results: a randomised, multicentre, open-label, phase 3b, non-inferiority study. Lancet. 2021 Dec 19;396(10267):1994-2005. doi: 10.1016/S0140-6736(20)32666-0. Epub 2020 Dec 9.

7. Figure 2. May be useful to give include identifiers for the 2 control macaques in the legend. Could also define eART (early ART) in the legend where the ART regimen is given.

As suggested, we added these two clarifications to the legend.

8. Figure 3. In the legend, include the time (usual) that the samples were obtained relative to dosing (e.g., predose) for panels A, B, C and D.

CAB was measured every week. RPV was measured 2-4 weeks after dosing. FTC-TP and TFV-DP were measured weekly 24h post-dose. We have added this information to the legend.

9. Suppl Table 1. I'm unsure about the appropriateness of calculating median values of CAB and RPV concentrations for the tissue groupings (eg, the various lymph nodes) in this table. Doing so seems to presume some expectation of comparability when it is known there are regional differences among these tissues with regard to drug penetration.

We agree and have deleted the median values.

Reviewer #3 (Remarks to the Author):

Daly et al. report on the efficacy of early treatment in SHIV-infected macaques using an ART combination that includes two long-acting antiretrovirals and the immune-stimulating agent VES (the latter in only one of the study arms). The post-treatment control of viral loads displayed by the treated macaques is quite impressive, suggesting that this drug combination could potentially induce a drastic remission, if not a cure, when adopted during the acute phase of infection.

The main drawback of the study is the lack of sufficient controls to determine the contribution of each treatment component to the observed remission, particularly since one of the drugs administered persisted in plasma throughout the entire post-therapy follow-up. Indeed, viral load control in macaques has recently been described upon treatment during acute infection without long-acting antiretrovirals (<https://pubmed.ncbi.nlm.nih.gov/38212337/>), although the mechanism of this control appears different from that described in the current study. Moreover, it is difficult to dissect the potential effects of the immunomodulatory treatment with VES, given that the duration of ART treatment was also altered in parallel to the addition of this compound.

Overall, despite these limitations, the study remains relevant and interesting due to the significant potential implications of the results obtained. Addressing the following points could help clarify the interpretation of some of the evidence presented:

We thank the reviewer for appreciating this work. We acknowledge that other mechanisms may contribute to remission and have now added the reference above in page 12 of the discussion.

1. Can any data be provided on rilpivirine persistence in tissues during the post-therapy follow-up?

This is an important question but unfortunately we don't have tissue drug level data.

2. Do the treated animals that displayed intact proviral DNA overlap with those displaying viral blips during follow-up or CD8 T cell depletion?

The answer to this interesting question is yes; macaque 10-47 which had a viral blip shortly after CD8 depletion also had intact proviral DNA. We now provide new IPDA data in Table 2 showing that this animal also had detectable intact proviruses 1 year after treatment cessation and 2 months after CD8 T cell depletion (or about 1 month before the transient virus blip). We describe this new observation in Page 9 of the results.

However, the overlap was not consistent. Macaque MAJ from the eART+VES group that showed a viral blip 5 months after treatment cessation had undetectable intact provirus at day 21 or at necropsy (Table 2).

3. It would be important to determine whether viral expression can be reactivated in vitro in the samples with detectable intact proviral DNA (e.g., using a viral outgrowth assay) after a washing period to reduce RPV concentration and in the presence of a strong reactivating stimulus (e.g., PMA/Ionomycin).

This is a very interesting experiment but unfortunately we don't have enough cells.

4. The contribution of VES to the observed effects is unclear. While adding control macaques could prove difficult, it would be relatively straightforward to test whether VES treatment was indeed associated with increased T-cell-mediated immune responses (e.g., through an ELISpot assay) or NK cell responses.

We agree with the reviewer in that these studies will help to better understand the role of VES

and plan to address this question in future studies.

Reviewer #4 (Remarks to the Author):

I co-reviewed this manuscript with one of the reviewers who provided the listed reports. This is part of the Nature Communications initiative to facilitate training in peer review and to provide appropriate recognition for Early Career Researchers who co-review manuscripts

We thank the reviewer for evaluating this manuscript.